# Development of a 'universal-reporter' outcome measure (UROM) for patient and healthcare professional completion: a mixed methods study demonstrating a novel concept for optimal questionnaire design

Rhiannon Macefield,[1,2] Sara Brookes,[1] Jane Blazeby,[1,2,3] Kerry Avery,[1,2,4] On behalf of the Bluebelle Study Group*

For numbered affiliations see end of article.

**Correspondence to**
Rhiannon Macefield;
R.Macefield@bristol.ac.uk

## ABSTRACT

**Objectives** To describe the novel concept of, and methods for developing, a 'universal-reporter' outcome measure (UROM); a single questionnaire for completion by patients and/or healthcare professionals (HCPs) when views on the same subject are required.

**Design** A mixed methods study with three phases—phase I: identification of relevant content domains from existing clinical tools, patient questionnaires and in-depth interviews with multistakeholders; phase II: item development using a novel approach that considered plain language in conjunction with medical terminology; and phase III: pretesting with multistakeholders using cognitive interviews.

**Setting** A case study in surgical wound assessment undertaken in two UK hospital trusts and one university setting.

**Participants** Patients who had recently undergone general abdominal surgery and healthcare professionals involved in post-surgical wound care.

**Results** Phase I: In the example case study, 19 relevant content domains were identified from two clinical tools, two patient questionnaires and 19 multistakeholder interviews (nine patients, 10 HCPs). Phase II: Domains were operationalised into items and subitems (secondary components to collect further information, if relevant). The version after pretesting had 16 items, five of which included further subitems. Plain language in conjunction with medical terminology was applicable in nine (27%) items/subitems. Phase III: Pretesting with 28 patients and 14 HCPs found that the UROM was acceptable to both respondent groups. An unanticipated secondary finding of the study was that the combined use of plain language and medical terminology during questionnaire development may be a useful, novel technique for evaluating item interpretation and thereby identifying items with inadequate content validity.

**Conclusion** UROMs are a novel approach to outcome assessment that are acceptable to both patients and HCPs. Combining plain language and medical terminology during item development is a recommended technique to improve

accuracy of item interpretation and content validity during questionnaire design. More work is needed to further validate this novel approach and explore the application of UROMs to other settings.

## BACKGROUND

Research often requires views from different stakeholders on the same subject. Reasons may be to combine different stakeholder responses to obtain comprehensive information to better answer the research question. Other reasons may be to compare stakeholder responses and explore any similarities or differences in perspectives, opinions or behaviours. Alternatively, there may be logistical reasons for obtaining views from different stakeholders to enable important data to be collected irrespective of who is available to provide it. In a clinical trial, for

example, the frequency and severity of symptoms and adverse events might be self-reported by the patient and/or judged by an observer, such as a healthcare professional (HCP).[1 2] In these situations, assessment tools and questionnaires are usually developed separately for use by specific stakeholder groups. Consequently, two different tools/questionnaires that intend to measure the same construct (concept) may use different terminology to suit the target audience. In the assessment of wounds for surgical site infection, for example, there are tools for clinical staff that use medical terminology such as 'purulent drainage' and 'spontaneous dehiscence'. Alternatively, separate patient-completed questionnaires use plain language descriptions asking patients, for example, about 'discharge or leakage of fluid' and whether the 'edges of the wound separated or gaped open'.

While separate stakeholder-specific tools may intend to measure the same constructs, uncertainty remains about whether this is achieved in practice. Evidence suggests that even minor alterations to item (question) wording can lead respondents to draw on different sources of information, subsequently affecting their responses.[3 4] It is likely, therefore, that the use of different stakeholder-specific terminology in separate tools for patients and for HCPs that intend to measure the same construct may introduce a degree of variation in the way that they are interpreted and subsequently the response that is provided. Specifically, differential understanding of items between individual respondents or stakeholder groups can compromise the measure's content validity or degree to which the content adequately reflects the construct being measured. This can have implications for drawing accurate conclusions when combining or comparing data collected from different stakeholder groups. Equitable interpretation of items by different respondents is therefore essential to ensure that the data collected by the separate measures are accurate and valid. It is hypothesised that developing a universal-reporter outcome measure (UROM) which uses a single set of terminology to collect data from either patients and/or HCPs may reduce variation in interpretation and thereby optimise the content validity of the measure. This study introduces the concept and method for developing a UROM, illustrated within a case study of surgical wound assessment.

## METHODS

### Case study: development of an outcome measure for surgical site infection (SSI)

The concept of a UROM originated as a solution to a problem within a feasibility study of surgical wound assessment. The feasibility study was performed to explore whether it would be possible to conduct a large randomised controlled trial (RCT) of different wound dressing strategies. The proposed primary outcome of the main RCT was surgical site infection (SSI) at 30 days post-surgery.[5] Assessment of SSI at this timepoint would typically be after the patient had been discharged from hospital and was recovering at home. At the time of the feasibility study, separate stakeholder-specific tools for evaluating surgical wounds to assess SSI were available for HCPs and for patients though they had several limitations.[6–10] The clinical tools for HCPs to complete, for example, were designed for use while patients were still in hospital, predominately used medical terminology and were complex to complete.[6 8] The questionnaires for patients were developed from a clinical perspective, did not involve patients in their development and had not been formally validated.[7 9] One aim of the feasibility study, therefore, was to develop and validate an outcome measure to assess wounds for infection that was suitable for use both in-hospital after the patient had been discharged. While it would have been possible to develop two separate stakeholder-specific measures, it was recognised that this may compromise the validity with which the construct of SSI was measured. A single UROM to evaluate surgical wounds, suitable for completion by patients or HCPs, was therefore developed.

### Development of the UROM

The UROM was developed using established methods for developing new outcome measures[11 12], adapted to address specific issues relevant to developing a 'universal' measure for patient and/or HCP completion (table 1). The following sections provide a brief description of these unique considerations and adapted methods, drawing on the surgical wound case study as an example. A full description of the development and evaluation of the surgical wound outcome measure (including assessments of reliability, a comparison of patient and HCP responses and clinical validity) has been published elsewhere[13 14] and is outside the scope of this paper.

**Table 1** Phases of UROM development

| | Established methods for outcome measure development | Adapted/novel methods relevant to UROM development |
|---|---|---|
| Phase I | Identification of content domains | Emphasis on using a multistakeholder perspective to identify domains of importance. |
| Phase II | Item construction | A multistakeholder approach considering plain language in conjunction with medical terminology. |
| Phase III | Pretesting and evaluation of content validity | Cognitive interviews with multiple stakeholders. |

UROM, universal-reporter outcome measure.

## Phase I: identification of content domains using a multistakeholder perspective

The first phase in the development of any new measurement instrument typically involves the identification of important content domains (ie, areas of interest potentially relevant to include in the new tool). Typical sources include the existing literature and interviews with key stakeholders to elicit expert opinion and experience.[12] The focus of the UROM development was to consider patient and HCP perspectives together to identify domains of importance to either or both stakeholders for consideration to include in a single, universal tool.

In the case study, health domains (defined as the sign, symptom or wound care intervention relevant to SSI assessment and management) important to patients and HCPs for possible inclusion in the new SSI measure were identified using existing guidelines for questionnaire development.[11 15] First, a content analysis of the two most commonly used existing clinical tools identified in a previous systematic review,[16] and their two associated patient questionnaires, was undertaken. In addition, in-depth interviews with 19 stakeholders were conducted; nine patients who had experience of wound infection and 10 HCPs involved in post-surgical care. Details of the existing tools, the methods for analysing their content and the interview sampling strategy and data collection have previously been reported.[13] Importantly for the UROM, data from the analysis of existing tools and interviews were combined to provide a list of all the domains considered to be relevant to SSI assessment, irrespective of whether the source was from a patient's and/or a HCP's perspective.

## Phase II: item construction using a multistakeholder approach; considering plain language in conjunction with medical terminology

The second phase in the development of a new measurement instrument usually involves the conversion or 'operationalisation' of domains (identified in phase I) into items for a questionnaire. For the UROM, a novel approach was applied to item construction; using both plain language and medical terminology wherever possible. The reason for doing this initially was to use language that was understood by, and was familiar to, each stakeholder group, to facilitate easy and timely completion of the outcome measure.

In the case study, the list of important SSI domains identified in phase I were considered for inclusion in the UROM. Domains considered to be unsuitable for patient report were excluded. Item construction was performed by four members of the case study team (JMB, RM, TM, BR), experts in the design and use of questionnaires including patient-reported outcome measures and professionals in the clinical field. First, plain language was used to describe the SSI domain in a clear and unambiguous way. Language was targeted for a lay audience without technical or medical terms, following standard recommendations.[12 15 17] Next, medical terminology (if

### Box 1 Example item showing plain language with medical terminology in parentheses

Was there redness spreading away from the wound? (erythema/cellulitis)

it existed) relating to the SSI domain was included in parentheses at the end of the item. An example is illustrated in box 1.

Response categories took the form of either a binary yes/no response or an ordinal scale (initially a five-point scale 'not at all', 'a little', 'moderately', 'quite a bit' and 'very much') as appropriate to the individual item. The remaining structure and layout of the questionnaire was designed to be simple, clear and straightforward for patients and/or HCPs to complete, in accordance with established guidelines. The UROM was produced in two formats. One format was a paper-copy questionnaire to post to patients after leaving hospital. The other format was a paper-copy case report form for HCPs to complete when conducting observer wound assessments either on the telephone or face to face as part of the wider feasibility study. Items and response categories were identical. The only minor difference was the use of first-person or third-person narrative, necessary for whether the tool was to be completed by a patient or an observer (box 2).

## Phase III: pretesting and evaluation of content validity: cognitive interviews with multiple stakeholders

The third phase in the development of any new measurement instrument involves pretesting with a sample of participants from the target population. Asking potential recipients to complete early drafts of a new measurement instrument is critical for testing understanding, interpretation and identifying potential problems with its completion and use. Cognitive interviews are a valuable technique used during pretesting to examine content validity and ensure that items are comprehended as intended.[12 18] Pretesting of the case study outcome measure has previously been described in detail.[13] Methods of specific relevance to development of the UROM are emphasised and expanded below.

### Participants and recruitment

In the case study, both patients and HCPs were invited to take part in cognitive interviews to pretest early versions of the measure. Patients were those who had recently undergone general abdominal surgery, identified and approached by research nurses/members of the study team in two UK hospital trusts. HCPs were those involved

### Box 2 Example item showing first-person and third-person narrative

Has your wound been drained? (drainage of pus/abscess)
Has the wound been drained? (drainage of pus/abscess)

in post-surgical care, identified from the same hospital trusts and from the authors' university institution. Written information describing the study in detail was provided to all participants. Contact details of interested participants were passed on to members of the study team and followed up by telephone or email to further discuss the study, answer questions and arrange an interview. Interviews were conducted by two researchers (RM and TM) between January and August 2015. Written consent was obtained prior to each interview.

### Cognitive interviews

Individual face-to-face cognitive interviews were conducted with participants to explore the overall acceptability, suitability and comprehensibility of the early versions of the UROM. The primary aim of the interviews was to examine the suitability of the UROM as an outcome measure for SSI. Specific objectives were to refine aspects of the questionnaire, including the layout, item phrasing, instructions and response categories. Additionally, and of specific relevance to development of a UROM, interviews explored views on items that included both plain language and medical terminology.

Participants were shown the questionnaire and asked to complete the items relating to their current experience (patients) or a recent or hypothetical patient case (HCPs). Participants were asked to vocalise their thoughts as they read and responded to each item using a 'think aloud' technique.[18] Completion of the questionnaire was observed by the researcher who then used probing questions to explore the participants' thoughts in more depth. Interpretation, accuracy and general opinions on the use of medical terminology alongside plain language in the questionnaire were sought. Question probes, for example, asking HCPs "Is (the plain language) a suitable description of the medical term?" were used to explore the accuracy of the item for measuring the intended underlying construct. Areas for investigation and specific items for discussion were identified and evolved throughout the course of interviews. Revisions were made to the provisional draft and new versions tested in subsequent interviews with new participants until findings indicated that no further revisions were required.

### Data analyses

Interviews were audio-recorded and written up in descriptive memoranda summarising key findings and suggestions for improvements to the questionnaire. Selected relevant quotations were transcribed verbatim. Interviews, analyses and modifications to the questionnaire were performed as an iterative process so that revisions to the questionnaire could be explored in subsequent interviews. Two researchers (RM and TM) independently conducted and summarised interviews, cross-checking approximately 25% of audio-recordings and memoranda to maximise rigour and reliability of the findings.[19]

**Table 2** Identified domains of importance for inclusion in the case study UROM for surgical wound assessment

| Domain relevant to SSI assessment | Existing tool | |
| --- | --- | --- |
| | Patient questionnaire(s) | Clinical tool(s) |
| 1. Wound healing | ✓ | x |
| 2. Wound heat | ✓ | ✓ |
| 3. Wound redness | ✓ | ✓ |
| 4. Wound discharge | ✓ | ✓ |
| 5. Layers separating—spontaneous | ✓ | ✓ |
| 6. Wound swelling | ✓ | ✓ |
| 7. Wound pain | ✓ | ✓ |
| 8. Fever | x | ✓ |
| 9. Contact with healthcare professional | ✓ | x |
| 10. Dressing needed | ✓ | x |
| 11. Antibiotics needed | ✓ | ✓ |
| 12. Layers separating—deliberate | x | ✓ |
| 13. Hospital admission | ✓ | x |
| 14. Drainage needed | x | ✓ |
| 15. Wound cleaning | ✓ | ✓ |
| 16. Abscess | ✓ | ✓ |
| 17. Microbiology | ✓ | ✓ |
| 18. Prolonged hospital stay | x | ✓ |
| 19. Smell* | x | x |

*Identified from stakeholder interviews.
SSI, surgical site infection; UROM, universal-reporter outcome measure.

### Patient and public involvement

Patients and members of the public were involved throughout the study. Two patient and public representatives were included on the study steering committee. A meeting was held with a group of patients to discuss the design and conduct of the study and to refine patient-facing study documents.

## RESULTS
### Phase I: identification of content domains using a multistakeholder perspective

In the case study, 19 relevant content domains (covering SSI signs, symptoms and wound care interventions) were identified from the existing tools and in-depth interviews (table 2). Of these 19 domains, 18 were identified from at least one of the existing clinical tools and/or patient questionnaires and were supported by interview data. One domain (smell) was not identified in any existing tools but was found to be important in interviews with both patients and HCPs.

## Phase II: item construction using a multistakeholder approach; considering plain language in conjunction with medical terminology

Seventeen of the 19 domains identified in phase I were developed into items for the first draft of the UROM. Two domains were excluded (microbiology and prolonged hospital stay) as they were considered unsuitable for patient report, and information could more reliably be obtained through other sources (eg, hospital records). All items were intended to be completed by all respondents, with some items having secondary components (subitems) to collect further information, where relevant. For example, if a participant responded to an item indicating that a symptom was present, further questions captured more details about that symptom.

The first draft of the UROM prior to pretesting included 13 items, of which six included secondary subitems to collect further information. Eight medical terms were included in parentheses after the plain language either in the items or secondary subitems. It was not applicable to include medical terminology in the remaining items because a medical description did not exist for the construct being addressed; for example, "Has the wound been smelly?". In the first draft, eight items had ordinal response categories and five items had binary yes/no response options. Responses of 'don't know' were also included to explore whether participants required this option and identify potentially problematic items to answer.

## Phase III: pretesting and evaluation of content validity: cognitive interviews with multiple stakeholders

Forty-two cognitive interviews (with 28 patients and 14 HCPs) were conducted. Participant characteristics and interview duration are summarised in table 3.

Detailed findings from the pretesting phase of the case study SSI outcome measure have previously been reported.[13] Findings of particular relevance to UROM design are described in detail below.

### Modifications to the UROM during pretesting

Throughout pretesting and the iterative process of interviews and revisions, the UROM was modified eight times. The final version after pretesting included 16 items for assessing SSI, with five having secondary subitems to collect further information. Nine medical terms were included in parentheses at the end of items/subitems (see online supplementary file 1).

General modifications (not specific to UROM design) included revision of the ordinal response categories from a five-point to a four-point scale ('not at all', 'a little', 'quite a bit' and 'a lot') because a middle category of 'moderately' was found to be uninformative. The filter question at the beginning of the measure ("Have you had any problems with the healing of your wound(s)?") was also dropped because data indicated that participants' answers to this filter question were often not concordant with their subsequent responses to

subsequent items (eg, participants responded that they had no problems with wound healing but went on to report experiencing symptoms of problems with wound healing). General changes also included restructuring some items, for example, changing some secondary subitems to standalone items to minimise errors and reduce missing data.

Several changes were made to the provisional measure of specific relevance to UROM design. Most changes related to the use of plain language in conjunction with medical terminology. For example, one medical term (calor) was dropped as interviews revealed it was not a term that was used in current practice. Another medical term (spontaneous dehiscence) was added to an item where previously no medical description had been considered. Detail explaining the reason for this is provided below. In another item, one term (dressing) that was initially included in the plain language description was later moved to the end of the item in parentheses because interviews revealed that it was less understood by a lay audience than initially expected.

**Table 3** Participant demographics and interview duration

| | Number of participants (total n=42) |
| --- | --- |
| Patients, n (%) | 28 (66.7) |
| HCPs, n (%) | 14 (33.3) |
| Age, years (%) | |
| 21–30 | 1 (2.4) |
| 31–40 | 9 (21.4) |
| 41–50 | 5 (11.9) |
| 51–60 | 9 (21.4) |
| >60 | 18 (42.9) |
| Male, n (%) | 21 (50.0) |
| Clinical expertise* | |
| General practitioner | 3 (21.4) |
| Hospital/Research nurse/midwife | 4 (28.6) |
| Practice/Community nurse/midwife | 3 (21.4) |
| Surgical trainee | 4 (28.6) |
| Surgery type,† n (%) | |
| Caesarean section | 3 (10.7) |
| Upper GI | 9 (32.1) |
| Lower GI | 10 (35.7) |
| Hernia repair | 6 (21.4) |
| Duration of interview, min | |
| Median (range) | 25 (13–52) |

*HCP participants only.
†Patient participants only.
GI, gastrointestinal; HCP, healthcare professional.

**Box 3    Acceptability of items combining plain language and medical terminology**

Participant: "I just skipped over it… I did say 'What's that?' but it didn't concern me because I could answer the question… I did make the comment of what [is that] but I didn't worry about it and I just went on to the next bit." Patient participant, 1107

Participant: "I was… you know… interested [in the medical terms]. I didn't look at all of it… um, a couple I thought was interesting because it was Latin. That's what I thought. And also spontaneous dehiscence… I thought, gosh… so yeah I found it quite interesting."

Interviewer: "Did you find them [medical terms] confusing?"

Participant: "No… For instance that first one… I don't think I even saw…" Patient participant, 1104

Participant: "If I was… on my own receiving this I am a bit of a google searcher so I would probably have looked them up." Patient participant, 2030

**Box 4    Improved understanding and interpretation of items**

Item: Was there redness spreading away from the edges of the wound? (erythema and cellulitis)

Participant: "In that first one [item], because I was describing the redness under the skin – more deeper redness, purple - when I read that first question, it was the fact that I had some idea of what erythema and cellulitis are… I thought, well it wasn't those…but ended up saying a little because of the redness… it probably was erythema… but I wasn't sure."

Interviewer: "And if we didn't have that erythema and cellulitis in there?…"

Participant: "Yeh, I would then have probably thought… that it was [asking about] that [bruising]… but because I recognised those [erythema and cellulitis]… I think I know more or less what those two things are." Patient participant, 1081

## Acceptability of UROMs and items combining plain language and medical terminology

In general, neither patients nor HCPs reported significant concerns with the inclusion of medical terminology alongside plain language within the same item. Some patients reported that they found the medical terms interesting and educational. Other patients reported that they found the plain language alone sufficient for comprehension and therefore ignored or did not notice the medical terms (box 3). Concerns that medical terms may cause patients anxiety was raised in interviews with HCPs, although this was not supported by data from interviews with patients. One patient referred to the use of the Internet to look up medical terms but did not express any concerns about doing so (box 3).

## Improved understanding and interpretation of items

An unanticipated secondary unexpected finding of the study was that the combined use of plain language and medical terminology may be a useful, novel technique for evaluating item understanding and improving item interpretation during the process of developing the measure. Findings from the pretesting interviews indicated that the inclusion of a medical term alongside plain language in an item directly affected the way that participants interpreted and subsequently responded to items, by facilitating their understanding of the item. One participant, for example, explained how the presence of the medical term improved their understanding of the item, thereby enabling them to respond more accurately. This participant reported that they would have interpreted the item differently, and therefore responded differently, had the medical term not been included (box 4).

## Identification of items with inadequate content validity

Directly related to the finding that the use of plain language and medical terminology may improve participants' understanding and interpretation of items, the combined use of plain language and medical terminology during item development was found to be a

useful technique to maximise the content validity of the outcome measure. The tandem use of medical terminology and plain language identified several items that were ambiguous or insufficiently reflected the construct that was intended to be measured. For example, it became apparent during interviews that some participants who read the plain language were interpreting an item differently to others who were also reading and understanding the medical terminology (box 5). In this example, quotes from patient participants demonstrated that the item was not being interpreted as intended, while quotes from the HCP participants indicated that the plain language description was not an adequate reflection of the medical terminology. This led to the item being modified to include more detail in the plain language description.

In addition to the finding that using both plain language and medical terminology from the outset of item development may improve the content validity of the outcome measure, there was some evidence to suggest that adding medical terminology to an item that had initially been written using only plain language may also maximise content validity. For example, interviews indicated that the plain language item "Have the edges of any part of the wound separated?" was not specific enough for measuring the intended construct (cases where the wound had spontaneously broken down or 'dehisced'). Specifically, the item was being interpreted too broadly by both patients and HCPs and was therefore interpreted to overlap with another item later intended to measure the deliberate separation of the wound edges by a doctor or nurse ("Has your wound been reopened by a doctor or nurse?"') (box 6). A medical term ('spontaneous dehiscence') was added to this item and the plain language revised to "Have the edges of any part of the wound separated on their own accord". Subsequent interviews with HCPs indicated that, had this medical term not been included, the plain language alone may not have been interpreted to include more serious cases of wound breakdown.

---

**Box 5  Improved content validity of the construct to be measured**

Item: Has your wound been cleaned out? (debridement of wound)

Participant: "[reading] 'Has your wound been cleaned out?'… Yes it has been cleaned out…with this little plastic thing of liquid… saline stuff… They squirt this liquid in… put it on some gauze." Patient participant, 1083

Participant: "I think… urm… when I had the staples taken out I think it was pretty standard practice for the nurse to just clean the wound before…. I don't know what she put on but it was a bit of cotton wool and she just rubbed… something." Patient participant, 1104

Participant: "To me… cleaned out and debridement… isn't the same thing. Cleaned out is washing with saline and debridement is picking… slough… like yellow stuff out… or cutting dead skin away or scabs." Healthcare professional participant, 3000

Participant: "When you say cleaning out of the wound do you just mean, like, getting some water?…That [debridement] actually, to me, involves cutting… debridement is when you actually remove by cutting… or scraping… some dead tissue. Cleaned out, to me, just implies… oh, um, that you just gave it a bit of a clean… I completely understand what debridement of the wound means but, to me, cleaned out is not the same." Healthcare professional participant, 1142

Modified item: Has your wound been cleaned out to remove any dead tissue? (debridement of wound) draft version 6.0 10/04/2015

## DISCUSSION

This article describes the novel concept of, and a method for developing, a UROM. This is a single questionnaire developed to measure a construct using data collected from either patients and/or HCPs by using a single set of terminology comprising both plain language and medical terminology. A UROM may be required for logistical reasons (as in the example case study) or it may be for other purposes when there is a need to combine or compare responses from different stakeholders. Development of a UROM includes established methods for developing new outcome measures,[11 12] uniquely adapted to address specific considerations and requirements of a UROM. These considerations include incorporating the views of all key stakeholders in all phases of UROM

**Box 6  Improved content validity of the construct to be measured**

Item: Have the edges of any part of the wound separated?

Participant: "So what does that one mean?… so… it is separated… because it's not stitched up"… The actual wound was left open because they couldn't stitch it up." Patient participant, 1079

Participant: "What does that [separated] mean—like cut or something? Got bigger?" Patient participant, 1076

Modified item: Have the edges of any part of the wound separated on their own accord? (spontaneous dehiscence) draft version 2.0 05/02/2015

development and a novel approach to item construction by combining plain language alongside medical terminology.

Illustrated within a case study of surgical wound assessment, the findings from this study indicate UROMs are acceptable for completion by both patients and HCPs and ready for further evaluation in future work. An unanticipated secondary finding of the study was that the combined use of plain language and medical terminology during questionnaire development may be a useful, novel technique for evaluating item interpretation and thereby identify items with inadequate content validity. Development and use of a UROM is recommended for studies where it is appropriate and beneficial to measure a construct using data collected from either patients and/or HCPs.

The concept of a UROM, with items that combine plain language and medical terminology, represents a different approach to outcome measurement where traditionally tools for patients and HCPs are developed separately and used separately. Guidelines for the development of measurement instruments usually advise against the use of clinical or technical jargon (eg, medical terminology), particularly when the general public are the intended recipients.[12 20] In general, guidance recommends not to use medical terminology in patient literature to avoid any difficulty in understanding.[20–22] This study shows, however, that the use of medical terminology alongside plain language during the development of a measurement instrument can be beneficial for making sure items are interpreted as intended and reflect the intended construct to be measured. No patients in this study reported concerns with this approach.

To our knowledge, this is the first study to introduce and examine the concept of a UROM. The use of this method for ensuring content validity may be applicable and beneficial in a wider context. Within our research institution, for example, we have undertaken studies developing core outcome sets (COSs) for trials in oesophageal, colorectal and bariatric surgery where the views of patients and HCPs on the same subject were required.[23–25] A UROM, with items written in plain language and medical terminology in parentheses where appropriate, was used to collect the opinions of both patients and HCPs and prioritise outcomes of importance.[23–25] This concept is now recommended to COS developers as one approach to consider for describing outcomes to stakeholder groups.[26] Other potential advantages of using UROMs rather than separate questionnaires for patients and HCPs include: (1) the need for a single study to develop the tool rather than separate studies for patient and HCPs measures; (2) a more streamlined and efficient way of collecting outcome data, with easier administration and reduced costs by using the same measure and (3) ease of data synthesis as data from multistakeholders can be readily combined. Further work to examine the applicability of UROMs to different settings would be beneficial.

This study has several strengths. It is the first study, to our knowledge, to describe a UROM; an outcome measure intentionally developed for patient and healthcare professional completion. One-to-one cognitive interviews with patients and HCPs also allowed for a detailed examination of item comprehension and acceptability of a single tool combining plain language and medical terminology in both stakeholder groups. UROMs are a novel concept and, currently, their evaluation is limited to the findings from this single case study. The potential advantages of the UROM design for improving content validity identified in this study was an unanticipated finding, however, and was not a primary focus of the case study interviews. The number of direct examples for its evaluation are, consequently, limited. The exact extent and nature to which medical terminology influences participants' responses warrants further investigation. In box 4, for instance, it is assumed that the respondent was clearer or more accurate with their response as a direct result of reading the medical term. The possibility that the medical term may have introduced uncertainty, 'noise' or measurement error was not formally explored. The detailed validation of the SSI outcome measure and the accuracy of the tool for assessing wound infection has been reported in full elsewhere[14], however, did not explore this possibility.

In summary, a novel approach to outcome assessment and development of a UROM is described. Findings have shown that combining plain language and medical terminology within items can improve content validity. It is a recommended technique for the development of outcome measures in other situations where information from both patients and HCPs is required. Further work is now needed to explore the applicability of UROMs in other settings within and outside the field of surgical research.

**Author affiliations**
[1]The MRC ConDuCT-II Hub for Trials Methodology Research, Bristol Medical School: Population Health Sciences, University of Bristol, Bristol, UK
[2]National Institute for Health Research Bristol Biomedical Research Centre, Bristol Centre for Surgical Research, Bristol Medical School: Population Health Sciences, University of Bristol, Bristol, UK
[3]Division of Surgery, University Hospitals Bristol NHS Foundation Trust, Bristol, UK
[4]Bristol Medical School, Population Health Sciences, University of Bristol, Bristol, UK

**Acknowledgements** The authors acknowledge Alexandra Nicholson and Tom Milne for their roles in conducting participant interviews for the development of the outcome measure. The research nurses and administrative staff within the Bristol Surgical Research team at University Hospitals Bristol NHS Foundation Trust and Clinical Research Nurses and administrative staff at Queen Elizabeth Hospital Birmingham for their roles in identifying and approaching patients, and staff within the Clinical Trials and Evaluation Unit, Bristol are also gratefully acknowledged for their roles in study administration. Recruitment and data collection for the Bluebelle study would not have been possible without the substantial contributions of the following surgical trainees from University Hospitals Birmingham NHS Foundation Trust, University Hospitals Bristol NHS Foundation Trust and North Bristol NHS Trust: Benjamin Waterhouse, Sean Strong, William Seligman, Lloyd Rickard, Samir Pathak, Anwar Owais, Jamie O'Callaghan, Stephen O'Brien, Dmitri Nepogodiev, Khaldoun Nadi, Charlotte Murkin, Tonia Munder, Tom Milne, David Messenger, Matthew Mason, Morwena Marshall, Jessica Lloyd, Jeffrey Lim, Kathryn Lee, Vijay Korwar, Daniel Hughes, George Hill, Mohammed Hamdan, Hannah Gould Brown, James Glasbey, Caroline Fryer, Simon Davey, David Cotton, Benjamin Byrne, Oliver Brown, Natalie Blencowe, Katarzyna Bera, Joanne Bennett, Richard Bamford, Danya Bakhbakhi, Muhammad Atif, Elizabeth Armstrong, Piriyankan Ananthavarathan.

**Collaborators** The Bluebelle Study Group consists of the following members: Bluebelle grant co-applicants: Lazaros Andronis (Division of Health Sciences, University of Warwick), Jane Blazeby (Bristol Medical School, University of Bristol; University Hospitals Bristol NHS Foundation Trust), Natalie Blencowe (Bristol Medical School, University of Bristol; University Hospitals Bristol NHS Foundation Trust), Melanie Calvert (Centre for Patient Reported Outcomes Research (CPROR), Institute of Applied Health Research, and NIHR, Birmingham Biomedical Research Centre, University of Birmingham), Joanna Coast (Bristol Medical School, University of Bristol), Tim Draycott (North Bristol NHS Trust), Jenny Donovan (Bristol Medical School, University of Bristol; NIHR Collaboration for Leadership in Applied Health Research and Care West at University Hospitals Bristol NHS Trust), Rachael Gooberman-Hill (Bristol Medical School, University of Bristol), Robert Longman (University Hospitals Bristol NHS Foundation Trust), Jonathan Mathers (Institute of Applied Health Research, University of Birmingham), Tom Pinkney (Academic Department of Surgery, Queen Elizabeth Hospital, University of Birmingham), Barnaby Reeves (Bristol Medical School, University of Bristol), Chris Rogers (Bristol Medical School, University of Bristol), Andrew Torrance (Institute of Applied Health Research, University of Birmingham), Mark Woodward (University Hospitals Bristol NHS Foundation Trust). Other members of the Bluebelle study group: Kate Ashton (Clinical Trials and Evaluation Unit, Bristol Trials Centre, Bristol Medical School, University of Bristol), Gemma Clayton (Clinical Trials and Evaluation Unit, Bristol Trials Centre, Bristol Medical School, University of Bristol), Madeleine Clout (Clinical Trials and Evaluation Unit, Bristol Trials Centre, Bristol Medical School, University of Bristol), Jo Dumville (School of Nursing, Midwifery & Social Work, University of Manchester), Daisy Elliott (Bristol Medical School, University of Bristol), Lucy Ellis (Clinical Trials and Evaluation Unit, Bristol Trials Centre, University of Bristol), Rosie Harris (Clinical Trials and Evaluation Unit, Bristol Trials Centre, Bristol Medical School, University of Bristol), Richard Lovegrove (Worcestershire Acute Hospitals NHS Trust), Christel McMullan (Institute of Applied Health Research, University of Birmingham), Helen van der Nelson (North Bristol NHS Trust), Caroline Pope (Clinical Trials and Evaluation Unit, Bristol Trials Centre, Bristol Medical School, University of Bristol), Anne Pullyblank (North Bristol NHS Trust), Leila Rooshenas (Bristol Medical School, University of Bristol), Dimitrios Siassakos (University College London Hospital), Sean Strong (Bristol Medical School, University of Bristol), Helen Talbot (Clinical Trials and Evaluation Unit, Bristol Trials Centre, Bristol Medical School, University of Bristol), Nicky Welton (Bristol Medical School, University of Bristol), Cathy Winter (North Bristol NHS Trust).

**Contributors** RM designed and pretested the measure, analysed and interpreted the cognitive interview data within the context of the wider case study and drafted the manuscript. SB was a major contributor in writing the manuscript. JB was the chief investigator for the Bluebelle study and was the major contributor for the concept of designing universal-reporter outcome measures. KA was a major contributor in writing the manuscript. All authors read and approved the final manuscript.

**Funding** This work was undertaken with the support of the Medical Research Council (MRC) ConDuCT-II Hub (Collaboration and innovation for Difficult and Complex randomised controlled Trials In Invasive procedures - MR/K025643/1) for Trials Methodology Research and the Royal College of Surgeons of England (RCS) Bristol Surgical Trials Centre. The Bluebelle study was funded by the National Institute for Health Research (NIHR) HTA Programme (project number 12/200/04). RM, JB and KA were supported by the NIHR Biomedical Research Centre (BRC) at the University Hospitals Bristol NHS Foundation Trust and the University of Bristol. The views expressed in this article are those of the authors and not necessarily those of the MRC, NHS, NIHR or the Department of Health and Social Care. JB holds an NIHR Senior Investigator award.

**Competing interests** None declared.

**Patient consent for publication** Not required.

**Ethics approval** Ethical approval was granted by the NHS Health Research Authority Research Ethics Committee London—Camden & Kings Cross (14/LO/0640) (Bluebelle study Phase A).

**Provenance and peer review** Not commissioned; externally peer reviewed.

**Data availability statement** All data relevant to the study are included in the article or uploaded as supplementary information.

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
