## [Reviewer comments · BMJ Open]

ARTICLE DETAILS

TITLE (PROVISIONAL)	Development of a 'universal-reporter' outcome measure (UROM) for patient and healthcare professional completion: a mixed methods study demonstrating a novel concept for optimal questionnaire design
AUTHORS	Macefield, Rhiannon; Brookes, Sara; Blazeby, Jane; Avery, Kerry

VERSION 1 - REVIEW

REVIEWER	Janeth Leksell School of Education, Health and Social sciences, Dalarna University, Sweden Medical Sciences, Uppsala University, Sweden
REVIEW RETURNED	01-Apr-2019

GENERAL COMMENTS	I do not agree with the authors conclusion: "Findings have shown that combining plain language and Medical terminology within items can improve content validity". As far as I can understand this was not the aim with the study. It is difficult to follow the step in the validation phase. The different phases are not transparently described. Perhaps a table could help the reader to understand what the authors have done. I could give an example: Phase 1 Item development Phase 2 Expert review and Evaluation of content validity Phase 3 Cognitive interviews ... and so on I lack a reference that clarifies which research method the authors have used in the development of the UROM. Perhaps, if the authors rewriting the article, it could be published. I think it is very important to develop patient reported outcome measurements.
--

REVIEWER	Jan van der Meulen London School of Hygiene and Tropical Medicine
REVIEW RETURNED	01-Apr-2019

GENERAL COMMENTS	This is a paper describing in detail the evaluation process of a single instrument to measure outcomes of surgical wound infection that can be used both by patients and healthcare professionals. I have a number of comments: First, I did not find a version of the final developed instrument included in the paper (or I must have missed it). It is for the reader
--

	easier to engage with the paper if he or she can look at the actual wording of the items and the response categories. Second, I'm not sure how the developers envisage that this questionnaire is going to be used. Is the idea of having a universal reported outcome measure that you can use it irrespective of whether it was completed by a patient or by a healthcare professional? Or is its main purpose that you can compare responses from the patients and healthcare professionals? Third, the core of the paper describes the development of the instrument. There is very little "evaluation". For example, a key question that I would have been interested in - highly relevant for assessing the value of the questionnaire - is the agreement or lack of it between patients and healthcare professionals. Fourth, there is an implicit assumption that it is helpful or beneficial that patients and healthcare professionals answer the same questions. You could argue that this is not always the case, for example with respect to pain or the extent to which a condition bothers a patient. Some further discussion about the specific purposes of a universal instrument are therefore very welcome. Fifth, and building on my third comment, I'd be very interested in seeing some actual results based on the application of this new instrument. To get an idea of its usefulness, it is helpful to assess how the responses are distributed across the response categories. Are the responses evenly distributed for all or most of the questions or are there questions with very unbalanced responses (i.e. most responses within one or two response categories)? Sixth, the text of the Abstract is very "generic", especially with respect to the Results paragraph. Would it be possible to provide more specific results, also focusing on wound assessment as the specific area of application?
--	---

VERSION 1 – AUTHOR RESPONSE

Reviewers' comments and responses:

Reviewer: 1

1. I do not agree with the authors conclusion: "Findings have shown that combining plain language and Medical terminology within items can improve content validity". As far as I can understand this was not the aim with the study.

Response

We apologise that the reasons for this conclusion were not clear. The finding that plain language and medical terminology combined within items improved content validity had not been anticipated a priori. The reason we used plain language and medical terminology in items initially was purely because we wanted to use terminology that would be familiar to both patients and healthcare professionals who would be completing the questionnaire. When conducting cognitive interviews to pre-test the questionnaire, however, we found that the combination of plain language and medical

terminology presented us with a useful technique for identifying inadequate content validity. For example, it showed us where there was ambiguity or insufficient detail in the wording of items, and highlighted items that needed to be rephrased to reflect the underlying issue as intended. It also showed us that item interpretation and understanding was improved when both plain language and medical terms were used.

These were beneficial findings that emerged from the data that we felt would be of interest to the research community and future questionnaire developers. In the paper, we report these findings and conclude that combining plain language and medical terminology may help to identify problems with content validity. We recommend that it is a useful technique that could be used by questionnaire developers in the future.

We have revised the wording of the abstract (results and conclusion) in the following ways, and have added a sentence to the methods section of the main text to make this clearer:

- i. Page 2, abstract: “An unanticipated secondary finding of the study was that the combined use of plain language and medical terminology during questionnaire development may be a useful, novel technique for evaluating item interpretation and thereby identifying items with inadequate content validity.”
 - ii. Page 2 abstract: “Combining plain language and medical terminology during item development is a recommended technique to improve accuracy of item interpretation and content validity during questionnaire design”
 - iii. Page 9, methods: “...a novel approach was applied to item construction, using both plain language and medical terminology where possible. The reason for doing this initially was to use language that was understood by, and familiar to, each stakeholder group, to facilitate easy and timely completion of the outcome measure.”
2. It is difficult to follow the step in the validation phase. The different phases are not transparently described. Perhaps a table could help the reader to understand what the authors have done. I could give an example: Phase 1 Item development Phase 2 Expert review and Evaluation of content validity Phase 3 Cognitive interviews ... and so on

Response

We thank the reviewer for this suggestion to make the phases of the study easier to follow. We have used clearer ways to describe the three phases distinctly in the abstract, methods and results, for example starting sentences with ‘Phase 1...’ and by revising the subheadings that describe the three different phases. We have also included a new table in the methods (page 8) as suggested by the reviewer to help make the three study phases clearer to understand:

Table 1. Phases of UROM development

	Established methods for outcome measure development	Adapted/novel methods relevant to UROM development
Phase 1	Identification of content domains	Emphasis on using a multi-stakeholder perspective to identify domains of importance
Phase 2	Item construction	A multi-stakeholder approach considering plain language in conjunction with medical terminology
Phase 3	Pre-testing and evaluation of content validity	Cognitive interviews with multiple stakeholders

We have revised the manuscript accordingly in the following places:

- i. Page 2, abstract: words “Phase 1)... Phase 2)... Phase 3)” added
- ii. Page 8, methods: words “Phases of development were: Phase 1)...; Phase 2)... and Phase 3)...” added
- iii. Pages 8, 9, 11 methods & 13, 14, 15 results: subheadings: “Phase 1) Identification of content domains using a multi-stakeholder perspective”

“Phase 2) Item construction using a multi-stakeholder approach; considering plain language in conjunction with medical terminology”

“Phase 3) Pre-testing and evaluation of content validity: cognitive interviews with multiple stakeholders”

3. I lack a reference that clarifies which research method the authors have used in the development of the UROM.

Response

We followed a multi-phase framework for development and early evaluation of new outcome measures. This framework was developed by the European Organisation for Research and Treatment of Cancer (EORTC) and is endorsed by the COSMIN (COnsensus-based Standards for the selection of health Measurement Instruments) initiative. In the original manuscript we included the following two references to this framework in the method). We hope these references are satisfactory examples of the research methods we used.

- Johnson, C., Aaronson, N., Blazeby, J., Bottomley, A., Fayers, P., Koller, M., et al. (2011). EORTC Quality of Life Group. Guidelines for developing questionnaire modules. (4th ed.). Brussels.
- de Vet, H. C. W., Terwee, C. B., Mokkink, L. B., & Knol, D. L. (2011). Measurement in medicine: a practical guide: Cambridge University Press.

Reviewer: 2

1. First, I did not find a version of the final developed instrument included in the paper (or I must have missed it). It is for the reader easier to engage with the paper if he or she can look at the actual wording of the items and the response categories.

Response

We thank the referee for making this suggestion. A version of the final developed instrument is now included as a supplementary file to help engage the reader, showing the wording of the items and the response categories.

The aim of the paper is to describe the concept of, and a method for developing, a universal-reported outcome measure (UROM) in general using the case study as an example. It does not intend to give a full description of the development and validation of the case study instrument. This has been described in detail elsewhere and the final developed instrument is included in that full report (Macefield et al., 2017). We do agree, however, that a version of the measure may be helpful to readers in the current paper and have now included it in the revised submission.

2. Second, I'm not sure how the developers envisage that this questionnaire is going to be used. Is the idea of having a universal reported outcome measure that you can use it irrespective of whether it was completed by a patient or by a healthcare professional? Or is its main purpose that you can compare responses from the patients and healthcare professionals?

Response

We envisage using a UROM for any purposes when the views of patients and healthcare professionals on the same subject are required. This may be for logistical reasons, for example, to collect data from either the patient or the healthcare professional. In these situations the UROM can be used irrespective of who is available to provide that data, and this was the purpose of the UROM in the case study in surgical wound assessment. In other studies and contexts, we envisage a UROM could be used to compare responses from patient or the healthcare professional if required. A UROM may also be used when there is a need to combine responses, for example, to get collective and comprehensive information to better answer a research question.

To explain this more clearly, we have revised the text in the introductory paragraph of the manuscript. We have also revised the first paragraph of the discussion. Details are as follows:

- i. Page 5, background: "Research often requires views from different stakeholders on the same subject. Reasons may be to combine different stakeholder responses, in order to obtain comprehensive information to better answer the research question. Other reasons may be to compare stakeholder responses and explore any similarities or differences in perspectives, opinions or behaviours. Alternatively, there may be logistical reasons for obtaining views from different stakeholders, to enable important data to be collected irrespective of who is available to provide it."
 - ii. Page 22, discussion: "This article describes the novel concept of, and a method for developing, a 'universal-reporter' outcome measure (UROM). This is a single questionnaire developed to measure a construct using data collected from either patients and/or HCPs by using a single set of terminology comprising both plain language and medical terminology. A UROM may be required for logistical reasons (as in the example case study) or it may be for other purposes when there is a need to combine or compare responses from different stakeholders."
3. Third, the core of the paper describes the development of the instrument. There is very little "evaluation". For example, a key question that I would have been interested in - highly relevant for assessing the value of the questionnaire - is the agreement or lack of it between patients and healthcare professionals.

Response

The paper does not intend to give a full evaluation of the case study measure, as this has been reported in detail and published elsewhere. The intention of the current paper is to introduce the reader to the general concept of a UROM, and to describe the specific methods for its development that are unique to its 'universal-reporter' purpose. For example, identifying content domains from a multi-stakeholder perspective and combining plain language and medical terminology in item development. It also intends to demonstrate the unexpected finding that the combination of plain language and medical terminology within items improved content validity, to highlight that it is a useful technique for future questionnaire developers. The paper uses the outcome measure for surgical wound assessment as an example case study to demonstrate these methods and findings. We agree that a comparison of patients' and healthcare professionals' responses is a very important aspect of the validation of the surgical wound outcome measure and this analysis was included in the separate publication.

We apologise that there was a lack of clarity and have revised the manuscript to address this, specifically the title and the objectives. We have also revised the methods to make it clear to the reader that full details describing the validation of the case study outcome measure are reported elsewhere. Revision are as follows:

- i.* Page 1, title: "Development and early evaluation of a 'universal-reporter' outcome measure (UROM) for patient and healthcare professional completion: a mixed methods study demonstrating a novel concept for optimal questionnaire design"
 - ii.* Page 2, abstract: "Objectives: To describe the novel concept of, and methods for developing, a 'universal-reporter' outcome measure (UROM)"
 - iii.* Page 7, methods: "The UROM was developed using established methods for developing new outcome measures adapted to address specific issues relevant to developing a UROM (Table 1). The following sections provide a brief description of these unique considerations and adapted methods, drawing on the surgical wound case study as an example. A full description of the development and evaluation of the surgical wound outcome measure (including assessments of reliability, a comparison of patient and HCP responses, and clinical validity) has been published elsewhere and is outside the scope of this paper."
4. Fourth, there is an implicit assumption that it is helpful or beneficial that patients and healthcare professionals answer the same questions. You could argue that this is not always the case, for example with respect to pain or the extent to which a condition bothers a patient. Some further discussion about the specific purposes of a universal instrument are therefore very welcome.

Response

We agree with the reviewer that it may not always be helpful or beneficial for patients and healthcare professionals to answer the same questions. The use of a UROM in these cases is not recommended. We envisage that a UROM could be used when it is appropriate for either or both stakeholders to answer the same question. This may be when an outcome of interest could be reported by either a patient or a healthcare professional, for example, an adverse event such as nausea or vomiting. We suggest that a UROM may be appropriate for logistical reasons, depending on who is available to provide the information. We also suggest that a UROM may be appropriate when the opinions of both patients and healthcare professionals are required on the same subject, for example, in a Delphi consensus survey.

We have revised the manuscript to describe examples where as UROM may be used, as highlighted in the response to comment 2 above. We have also added the following discussion point to the discussion:

- i.* Page 21, discussion: "Development and use of UROMs are recommended for studies where it is appropriate and beneficial to measure a construct using data collected from either patients and/or HCPs"
5. Fifth, and building on my third comment, I'd be very interested in seeing some actual results based on the application of this new instrument. To get an idea of its usefulness, it is helpful to assess how the responses are distributed across the response categories. Are the responses evenly distributed for all or most of the questions or are there questions with very unbalanced responses (i.e. most responses within one or two response categories)

Response

We apologise that it was unclear that these results were not intended to be reported in the current paper, and we refer the reviewer to our response to comment 3 above. The aim of the paper is to introduce the reader to the general concept of a UROM, and to describe the specific methods for its development that are unique to its 'universal-reporter' purpose. A detailed report of the validation study for the surgical wound measure is published elsewhere. It includes results such as the distribution of responses across the response categories, as well as other validation assessments including tests for reliability, a comparison of patient and HCP responses, and clinical specificity and sensitivity.

6. Sixth, the text of the Abstract is very "generic", especially with respect to the Results paragraph. Would it be possible to provide more specific results, also focusing on wound assessment as the specific area of application?

Response

As explained in the previous responses, the abstract is intended to be generic to describe the concept of a UROM in general, and methods for its development. Specific details and a full report of the development and validation of the case study outcome measure have been published elsewhere.

We have made revisions to the manuscript to make the objectives of the current paper more transparent and we hope this is now clearer to the reviewer and future readers.